# Two Retrotransposon Elements in Intron of Porcine *BMPR1B* Is Associated with Phenotypic Variation

**DOI:** 10.3390/life12101650

**Published:** 2022-10-20

**Authors:** Chenglin Chi, Jia He, Zhanyu Du, Yao Zheng, Enrico D’Alessandro, Cai Chen, Ali Shoaib Moawad, Emmanuel Asare, Chengyi Song, Xiaoyan Wang

**Affiliations:** 1College of Animal Science & Technology, Yangzhou University, Yangzhou 225009, China; 2Department of Veterinary Science, Division of Animal Production, University of Messina, 98168 Messina, Italy; 3Department of Animal Production, Faculty of Agriculture, Kafrelsheikh University, Kafrelsheikh 33516, Egypt

**Keywords:** pig, RIPs, *BMPR1B*, phenotype, repressor

## Abstract

It has been established that through binding to bone morphogenetic proteins (*BMPs*), bone morphogenetic protein receptor I B (*BMPR1B*) can mediate transforming growth factor β (TGF-β) signal transduction, and is involved in the regulation of several biological processes, such as bone and muscle formation and homeostasis, as well as folliculogenesis. Also known as FecB, *BMPR1B* has been reported as the major gene for sheep prolificacy. A number of previous studies have analyzed the relationship between single nucleotide polymorphisms (SNPs) in this gene and its related performance. In recent years, with the illustration of the effect of retrotransposon insertion on the expression of the proximal genes or phenotypic variation, retrotransposon insertion polymorphisms (RIPs) have been used as a novel type of molecular marker in the evaluation of evolution, population structure and breeding of plant and domestic animals. In this study, the RIPs in porcine *BMPR1B* gene were excavated, and thereafter verified using a comparative genome and polymerase chain reaction (PCR). The potential effects of phenotype, gene expression and functions related to RIPs were also explored. The results showed that 13 distinct RIPs were identified in introns of porcine *BMPR1B*. Among these, only *BMPR1B*-SINE-RIP9 and *BMPR1B*-LINE-RIP13 displayed a close relationship with the growth traits of Large White pigs. Moreover, the total number of *BMPR1B*-SINE^+/+^-RIP9 individuals born was found to be significantly higher than that of SINE^−/−^ (*p* < 0.05). These two RIPs showed an obvious distribution pattern among Chinese indigenous breeds and Western commercial breeds. The expression of *BMPR1B* in ovaries of adult *BMPR1B*-SINE^+/+^-RIP9 Sushan pigs was found to be significantly higher in comparison to those of *BMPR1B*-SINE^−/−^-RIP9 (*p* < 0.05). SINE insertion of *BMPR1B*-SINE-RIP9 and LINE insertion of *BMPR1B*-LINE-RIP13 were observed to significantly increase the activity of Octamer binding transcription factor 4 (OCT4) minipromoter in CHO and C2C12 cells (*p* < 0.01). Therefore, these two RIPs could serve as useful molecular markers for modulating the growth or reproductive traits in assisted selection of pig breeding, while the mechanisms of the insertion function should be studied further.

## 1. Introduction

Bone morphogenetic protein receptor I B (*BMPR1B*) is a member of the bone morphogenic protein (*BMP*) receptor family, which can effectively bind *BMPs* to mediate transforming growth factor β (TGF-β)-induced signal transduction events [1], and is involved in the regulation of important biological processes, such as bone and muscle formation and homeostasis [2,3], folliculogenesis [4], etc. The protein encoded by *BMPR1B* gene contains four conserved residues, which can play different roles in defining BMP2, 6, 7 and GDF5 ligand affinity [5]. The variation and expression change of the *BMPR1B* have been related to the human ovarian insufficiency and defects of folliculogenesis [6,7], Pierre Robin sequence [8], isolated acromesomelic dysplasia [9], neuroblastoma [10], growth plate defects [11] and cancers [2].

For domestic animals, the point mutation of A746G in the coding region of sheep *BMPR1B*, also known as FecB, was reported to cause a significant increase in the ovulation rate and Booroola phenotype in 2001 [12,13,14]. Since then, a number of studies have been performed to decipher the relationship between this causal mutation and the prolificacy of sheep or goats [15,16,17]. The expression difference among these mutation allele individuals has been identified, in order to further elucidate the mechanism of the A746G mutation on biological processes [18,19,20]. The gene editing technology on this point mutation was used to successfully obtain the FecB-mutated sheep [21]. However, the transgenic pigs generated by introducing 746G in *BMPR1B* showed a normal litter size performance [22]. In addition, other point mutation and indels in the *BMPR1B* gene were found to significantly affect the litter size of sheep [23,24]. Overall, this gene has been thought as the major gene involved in regulating sheep prolificacy, and conformed connecting with pig prolificacy [25]. Up to now, there have only been few studies reported on the retrotransposon insertion polymorphisms (RIPs) of *BMPR1B* in pig breeding.

Retrotransposons, which are the dominant type of transposable elements (TEs), can effectively contribute to one-third to one-half of the mammalian genome [26]. They are an interesting genome component that can mobilize and generate new copies of themselves, as they are reversely transcribed into the genome using the pattern of “copy and paste”. The cis-regulatory sequences contained in TEs could recruit the host-encoded factor, such as transcription factors (TFs), and ensure their amplification within the new loci of host [27]. Therefore, TE insertion can affect the proximal genes by functioning as enhancers, promoters, silencers and boundary elements contributing to the genome architecture [28,29].

According to the presence of long terminal repeats (LTRs) or non-LTR, retrotransposons can be further classified into two types. Endogenous retroviruses (ERVs) are the main part of LTRs. Non-LTR retrotransposons have been divided into long interspersed elements (LINEs) and short-interspersed elements (SINEs). Based on the retrotransposon insertion polymorphism, several types of molecular markers have been developed in plant breeding [30,31]. Recently, RIPs were also identified as a causative mutation of the phenotype, or closely related to the performance of domestic animals [32,33,34,35,36,37,38,39]. Considering the importance of the *BMPR1B* protein in several biological process, the RIPs in *BMPR1B* were excavated and verified. The potential relationship of these RIPs with pig growth, as well as the reproductive performance and the mechanism of these insertions, were identified to find the effective and novel molecular markers that could be used in pig breeding.

## 2. Materials and Methods

### 2.1. Animals and DNA Isolation

Ear or blood samples were collected from the twelve pig breeds or population including Duroc, Landrace, Large White, Sujiang, Diannan small ear, Erhualian, Wuzhishan, Bama, Tibetan, Meishan, Fengjing and wild boars. Genomic DNA was extracted using a standard phenol-chloroform extraction procedure and stored at −20 °C. DNA pool from 12 breeds or population were prepared for RIPs verification, with 6 individuals for each breed. A certain number of breeds, including Duroc, Landrace, Large White, Erhualian, Wuzhishan, Sushan and Sujiang were prepared for analyzing the population diversity. The correlation analysis was performed between the different RIP genotypes and the performance records of thickness of backfat, age at 30 kg bodyweight, age at 100 kg bodyweight, first litter size, number born alive, litter weight of Large White pigs. Both the origin of breeds and number have been shown in Appendix A. Among them, Large White, Duroc and Landrace are three lean type commercial breeds. Fengjing, Meishan, Erhualian and Wuzhishan are four fat type indigenous breeds. Sujiang is a cross breed which originated from Duroc, Jiangquhai and Fengjing. Sushan is also a cross breed developed from Meishan, Erhualian, Landrace and Large White in Jiangsu Province [40].

### 2.2. RIPs Excavation

The sequence of *BMPR1B* (Gene ID: 396691) with the flanking region (5 kb 5′ upstream and 3 kb 3′ downstream) were downloaded from NCBI (https://ncbi.nlm.nih.gov/, accessed on 17 May 2021). As the bait, *BMPR1B* sequences from the fourteen assembled non-reference genomes (Bama, Pietrain, Jinhua, cross-breed of Yorkshire/Landrace/Duroc, Landrace, Hampshire, Wuzhishan, Berkshire, Large White, Tibetan, Meishan, Bamei, Gottingen and Rongchang) were downloaded from the NCBI database (https://blast.ncbi.nlm.nih.gov/Blast.cgi, accessed on 19 May 2021). All the sequences were aligned to excavate structural variations using the ClustalX (Version 2.0). In addition, large structural variations (more than 50 bp and presenting more than one population) were retained for the retrotransposon annotation using RepeatMasker (Versions 4.0.7). The structural variations with over 60% similarity to the sequences in RepeatMasker were considered as the predicted retrotransposons and used for further identification by PCR.

The conservatism of *BMPR1B* genes from the different species, including pig (ENSSSCG00000029621), Chacoan Peccary (ENSCWAG00000015161), human (ENSG00000138696), cattle (ENSBTAG00000002081) and horse (ENSECAG00000012140), were analyzed using mVISTA (http://genome.lbl.gov/vista/mvista/submit.shtml, accessed on 21 May 2021).

### 2.3. RIPs Identification by PCR

The primers spanning the predicted retrotransposons were designed using Oligo7. The primer sequences and annealing temperature are shown in Appendix A. All the primers in this study were synthesized by Beijing Liuhe Huada Gene Technology Co. Ltd. (Beijing, China). PCR was carried out using ABI PRISM 3100 Genetic Analyzer (Applied Biosystems, Thermo Fisher Scientific, Waltham, MA, USA), and the reaction system was prepared as described previously by using the 12 DNA pools as template [38].

### 2.4. Population Diversity Based on RIPs

*BMPR1B*-SINE-RIP9 and *BMPR1B*-LINE-RIP13 were used to illustrate the genetic diversity of the represented breeds, including the lean-type commercial breeds (Large white, Landrace and Duroc), fat-type indigenous breeds (Fengjing, Meishan, Erhualian and Wuzhishan) and Jiangsu cross breeds (Sujiang and Sushan). Genotype frequency, allele frequency and Hardy-Weinberg test of RIPs were analyzed by POPGENE 1.32 [41]. Polymorphic information content (PIC) was calculated by the following formula:$$PIC=1-\sum_{i=1}^{m}P_{i}^{2}-\sum_{i=1}^{m-1}\sum_{j=i+1}^{m}2P_{i}^{2}P_{j}^{2}$$

### 2.5. qPCR

The primers (Appendix A) for the pig *BMPR1B* (mRNA XM_021100423.1) and *GAPDH* (mRNA XM_003126531.5) were designed based on the sequences available in GenBank Using Oligo7 (accessed on 3 June 2021). 

Four Sushan individuals for each genotype of *BMPR1B*-SINE-RIP9 and for LINE^+/+^ and LINE^+/−^ individuals of *BMPR1B*-LINE-RIP13 were slaughtered to obtain the ovary, backfat and longissimus dorsi. Thereafter, the tissue RNA was extracted using TRIzol (Tiangen, Beijing, China), and the quantity and quality of total RNA were monitored using a spectrophotometer (NanoPhotometer N60 Touch, Implen Gmbh, Munich, Germany). The cDNA was prepared using Takara Kit (Takara, Dalian, China). A real-time quantitative polymerase chain reaction (qPCR) was performed in a 25μL reaction mixture, containing 12.5 μL TB Green^®^ premix Ex TaqTM II (2×, Biomedical, Dalian, China), 1 μL each forward and reverse primer (10 ng/µL), 8.5 µL ddH2O, and 2 µL cDNA (100 ng/µL). The reaction conditions used were as follows: pre-denaturation at 95 °C for 30 s; denaturation at 95 °C for 3 s; and extension at 60 °C for 30 s (40 cycles). The relative expression levels of *BMPR1B* mRNA were analyzed using the 2^−ΔΔCt^ method, and the *GAPDH* gene was used as the internal reference.

### 2.6. Dual-Luciferase Reporter Assay

The insertion fragments with the double restriction enzyme cutting site of *BMPR1B*-SINE-RIP9 (273 bp) and *BMPR1B*-LINE-RIP13 (804 bp) were cloned from the Sushan genomic DNA (the primers have been listed in Appendix A), and verified by sequencing. The fragments were then inserted into the pGL3-basic vectors (Promega, Madison, WI, USA). Furthermore, Oct4 mini promoters were cloned from the pTol2-Oct4-mCherry vector [42] and inserted into pGL3-basic vectors with or without the insertion fragment to evaluate the potential enhancer or depressor activity. The correct recombinant plasmids were designated as pGL3-SINE-Oct4-basic and pGL3-LINE-Oct4-basic. Using a Lipofectamine 3000 reagent (Invitrogen, Carlsbad, CA, USA), 2 × 10^4^ CHO and C2C12 cells were plated and transfected with the plasmids. The cells were collected after 48 h to evaluate the luciferase activity using the dual luciferase reporter system (Promega, Madison, WI, USA) according to the manufacturer’s instructions. The luciferase activity was calculated by the following formula:$$\mathrm{Luciferase}\;\mathrm{activity}=\frac{\mathrm{Firefly}\;\mathrm{luciferase}\;\mathrm{activity}}{\mathrm{Renilla}\;\mathrm{luciferase}\;\mathrm{activity}}$$

### 2.7. Statistical Analysis

All the data has been shown as mean ± standard error, and *p* < 0.05 was considered a significant difference. The results were processed by SPSS17.0 (SPSS, Chicago, IL, USA) using one-way analysis of variance with Duncan’s test. The association analysis was also calculated by using one-way ANOVA in the SPSS software. The general linear model was used to determine the possible association between RIPs and growth and reproductive traits. Equation 1 for growth trait and 2 for reproductive traits used were as follows: (1) y_ijk_= u + g_i_ + a_j_ + s_k_ + e_ijk_, (2) y_ijk_ = u + g_i_ + s_k_ + e_ik_, where y_ijk_ was the phenotypic observation, u was the mean, g_i_ was the fixed effect of genotype, a_j_ was the fixed effect of gender, s^k^ was the fixed effect of born weight, and e_ijk_ or e_ik_ was the residual effect.

### 2.8. Animal Welfare 

All the treatments and protocols involving animals in this study were strictly performed in accordance with the guidelines of the Animal Experiment Ethics Committee of Yangzhou University (approval number: YZUDWSY2018-12).

## 3. Results

### 3.1. Thirteen RIPs Were Verified by PCR in the Pig BMPR1B Gene

By RepeatMasker annotation, 38 SINE, 17 LINE and 6 LTR insertion polymorphisms were predicted in the *BMPR1B* gene among the reference genome, and another 15 non-reference genomes. The PCR was employed to identify these predicted RIPs by using the specific primers (Appendix A), and 13 RIPs were verified in DNA pool (Figure 1A). They were all distributed in the introns of *BMRP1B* gene (Figure 1B). Most of the insertion loci were relatively not conserved (Figure 1C). The details, including the insertion sequence size, direction and subfamily of retrotransposon, have been listed in Table 1. These RIPs were named as *BMPR1B*-SINE-RIP1, *BMPR1B*-SINE-RIP2, *BMPR1B*-SINE-RIP3, *BMPR1B*-SINE-RIP4, *BMPR1B*-SINE-RIP5, *BMPR1B*-SINE-RIP6, *BMPR1B*-SINE-RIP7, *BMPR1B*-ERV-RIP8, *BMPR1B*-SINE-RIP9, *BMPR1B*-SINE-RIP10, *BMPR1B*-LINE-RIP11, *BMPR1B*-ERV-RIP12 and *BMPR1B*-LINE-RIP13, which corresponded to labels 1–13 in Figure 1B.

### 3.2. Association between RIPs and Phenotype of Large White Pigs

The association of 13 RIPs with the phenotype of Large White pigs was analyzed, and *BMPR1B*-SINE-RIP9 and *BMPR1B*-LINE-RIP13 were found to be closely associated with the growth and reproductive traits of Large White pigs. As shown in Table 2, the age at 100 kg body weight of SINE^−/−^ individuals for *BMPR1B*-SINE-RIP9 was observed to be significantly lower than those of SINE^+/−^ individuals (*p* < 0.01). The age at 30 kg body weight and 100 kg body weight of LINE^–/–^ individuals for *BMPR1B*-LINE-RIP13 was noted to be significantly lower than those of LINE^+/+^ individuals (*p* < 0.05). The first litter size of SINE^+/+^ individuals for *BMPR1B*-SINE-RIP9 of Large White pigs was significantly higher as compared to those of SINE^−/−^ individuals (*p* < 0.05) (Table 3).

### 3.3. BMPR1B-SINE-RIP9 and BMPR1B-LINE-RIP13 Distribution in the Different Pig Breeds

The potential distribution and genetic parameters of *BMPR1B*-SINE-RIP9 and *BMPR1B*-LINE-RIP13 in lean-type commercial breeds (Large White, Landrace and Duroc), fat-type indigenous breeds (Fengjing, Erhualian, Meishan, Jinhua and Wuzhishan) and cross breeds in Jiangsu province (Sujiang and Sushan) are shown in Table 4. The SINE^+^ frequency of *BMPR1B*-SINE-RIP9 was dominant in the lean-type breeds (Large White and Landrace) and cross breeds (Sushan and Sujiang) while the SINE^−^ was dominant in fat-type breeds (Wuzhishan). The LINE^+^ frequency of *BMPR1B*-LINE-RIP13 was obviously found to be higher in lean-type breeds (Duroc, Large White and Landrace) in comparison to cross breeds (Sushan and Sujiang) and fat-type breeds (Jinhua and Erhualian). Except for Large White pigs, all RIPs in the different breeds were in Hardy-Weinberg equilibrium. *BMPR1B*-SINE-RIP9 in fat-type indigenous breeds (Wuzhishan) and Sujiang were low polymorphic, while lean-type breeds (Duroc, Large White and Landrace) and Sushan were medium polymorphic. *BMPR1B*-LINE-RIP13 in the cross pigs (Sushan and Sujiang) and fat-type breeds (Jinhua and Erhualian) were low polymorphic, but in lean-type breeds (Duroc, Large White and Landrace) they were medium polymorphic.

### 3.4. The Expression Pattern of Pig BMPR1B in Sushan Pigs for Two RIPs

The expression in ovary, longissimus dorsi and backfat of the different genotype individuals of six-month old Sushan pigs for *BMPR1B*-SINE-RIP9 and *BMPR1B*-LINE-RIP13 was shown in Figure 2. For *BMPR1B*-SINE-RIP9, there was significant difference in the expression of *BMPR1B* among SINE^+/+^ and SINE^−/−^ individuals in ovaries (*p* < 0.05). Two genotypes, LINE^+/−^ and LINE^−/−^, were found in Sushan pigs for *BMPR1B*-LINE-RIP13. There was no significant difference noted between these two genotype individuals for the expression of *BMPR1B* in the three tissues of Sushan pigs (*p* > 0.05).

### 3.5. SINE and LINE Insertion Might Act as an Enhancer in CHO and C2C12 Cells 

The insertion fragments, which were 273 bp and 692 bp for *BMPR1B*-SINE-RIP9 and *BMPR1B*-LINE-RIP13, respectively, were amplified from Large White genomic DNA and then inserted into the pGL3-Oct4-basic vector. The recombinant plasmids were named as *BMPR1B*-SINE9-Oct4-Luc^+^ and *BMPR1B*-LINE13-Oct4-Luc^+^ (Figure 3C). These two plasmids were transfected into CHO and C2C12 cells, and the pGL3-Oct4-basic plasmid was used as a negative control (Figure 3D). The luciferase activity of *BMPR1B*-SINE9-Oct4-Luc^+^ and *BMPR1B*-LINE13-Oct4-Luc^+^ was found to be significantly higher than that of pGL3-Oct4-basic negative control (*p* < 0.01) in CHO and C2C12, Therefore, the 273 bp fragment and 692 bp fragment might act as an enhancer to affect the Oct4 promoter activity in CHO and C2C12 cells.

## 4. Discussion

*BMPR1B* is a type 1B receptor and a pivotal member of the BMP receptor family. The protein encoded by the *BMPR1B* gene mainly participates in the regulation of the BMP signaling pathway. BMPs are secreted ligands in the TGFβ superfamily, and can perform various important roles, including neurogenesis, organogenesis, pregnancy, cancer, etc [43]. All porcine *BMP* genes were screened to find out useful RIPs, but nothing was found within the gene and upstream or downstream flanking in the previous reports. The similar results were also observed in *GH*, *Leptin* gene, which is necessary for maintaining the different physiological processes including growth, development and reproduction. The studies have shown that 85–90% of mouse and human protein coding genes contain transposon elements sequences in their introns [44]. Thus, it is possible that essential genes might have developed measures to control the retrotransposon mobility in order to ensure proper physiological function [45]. However, their receptors could present some RIPs, which lead to the sequence diversification of organisms. All identified RIPs were found in the intron of the porcine *BMPR1B* gene. In our previous study, only 0.63% of retrotransposons were inserted into the exon of protein coding genes, while 35.1% retrotransposons were inserted into the intron of protein coding genes [46]. The considerable transposon elements’ under-representation in the coding regions could be attributed to a strong negative selection to avoid repercussions in the protein structure [47].

A number of studies have reported that *BMPR1B* is the major gene for determining the litter size of sheep. As a new-type molecular marker, 13 RIPs were used to analyze the correlation with the growth and reproductive traits of Large White pigs in this study. It was found that only *BMPR1B*-SINE-RIP9 and *BMPR1B*-LINE-RIP13 were closely related to the growth traits of Large White pigs, while *BMPR1B*-SINE-RIP9 was closely linked to the first litter size of Large White pigs. Using high throughput sequencing technology, it has been reported that the selective sweep that included *BMPR1B* (chr8:134.05–134.15 Mb and chr8: 123.3–147.4 Mb), which overlapped with the QTL to *BMPR1B*, significantly affected the number of piglets born [25,48]. *BMPR1B*-SINE-RIP9 and *BMPR1B*-LINE-RIP13 were within or very close to the selective sweep for litter size on the Chr8. Thereafter, the distribution of these two RIPs were evaluated in the representative pigs of three different type pigs, including fat-type, lean-type and cross breeds. Chinese indigenous pigs are famous for their high prolificacy, good meat quality, slow growth rate and low lean percentage in comparison with commercial pigs originating from Western nations [49,50,51,52,53]. There were obvious different distribution patterns noticed among these three types of pigs. Some nucleotide variants in the intragenic and intergenic regions indicated population characteristics among the different originated pig breeds [54,55,56,57]. *BMPR1B*-SINE-RIP9 and *BMPR1B*-LINE-RIP13 showed the obvious variable distribution pattern among the different type of breeds, which may be attributed to the difference of artificial selection concerned with these pigs. Therefore, we focused our attention on these two RIPs.

*BMPR1B* defects in mice have been shown to cause infertility [58], and high expression of *BMPR1B* in ovaries has been related to highly prolific sheep and goats [20,59]. In this study, the SINE-insertion individuals exhibited higher expression of *BMPR1B* in ovary and greater first litter size in comparison to individuals without SINE insertion. The function of *BMPR1B* was diverse, and the protein contributed to the growth and development of the mammal. The two RIPs were also found to regulate the growth rate of Large White pigs. However, the expression difference was not found in back fat and muscle among these genotypes. It is possible that the expression of *BMPR1B* in the tissues of brain was relatively high compared with other tissues, including the ovary [20,23], which made it play a major role through modulating the hypothalamic–pituitary axis to affect the growth and development. 

To explore the potential mechanism of retrotransposon insertion affecting the expression of *BMPR1B*, the dual-luciferase reporter assay was employed to evaluate the potential activities of insertion sequences. Retrotransposon can disperse vast amounts of promoters, enhancers, and even repressive elements [60,61]. In this study, the SINE and LINE insertion significantly increased the dual-luciferase activity in CHO and C2C12 cells. Since SINE insert is present upstream of CDS domain, it could act as an enhancer to increase the expression of *BMPR1B* in ovaries and affect the reproductive traits of Large White pigs. However, the mechanisms through which *BMPR1B*-LINE-RIP13 can affect growth traits is still unclear, but the function of LINE insertion should be further explored. Overall, combined analysis of the relationship between these two RIPs and related traits of Large White pigs could lead to useful molecular markers for assisted selection in pig breeding.

## 5. Conclusions

Thirteen distinct RIPs were identified in introns of porcine *BMPR1B* by comparative genome and PCR. Only *BMPR1B*-SINE-RIP9 and *BMPR1B*-LINE-RIP13 displayed a close relationship with the growth rate or first litter size of Large White pigs. These two RIPs exhibited an obvious distribution pattern that the SINE^+^ frequency of *BMPR1B*-SINE-RIP9 and the LINE^+^ frequency of *BMPR1B*-LINE-RIP13 was dominant in the commercial breeds in comparison to Chinese indigenous breeds, which may be attributed to the difference of artificial selection concerned with these pigs. The expression of *BMPR1B* in ovaries was altered significantly by the SINE insertion of *BMPR1B*-SINE-RIP9 in adult Sushan pigs (*p* < 0.05). SINE insertion of *BMPR1B*-SINE-RIP9 and LINE insertion of *BMPR1B*-LINE-RIP13 significantly increased the activity of OCT4 minipromoter in both CHO and C2C12 cells (*p* < 0.01). Therefore, these two RIPs could serve as useful molecular markers for predicting the growth or reproductive traits in the assisted selection of pig breeding. 

## Figures and Tables

**Figure 1 life-12-01650-f001:**
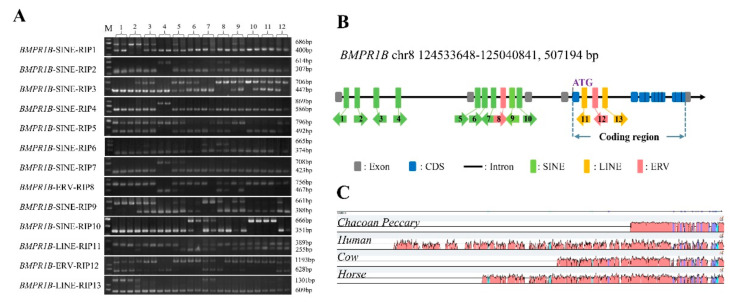
RIPs identification and distribution in *BMPR1B* gene. (**A**). RIPs identification by PCR and electrophoresis in DNA pool of twelve population. M: DL2000 marker; 1: Duroc, 2: Landrace, 3: Large White, 4: Bama, 5: Diannan small-ear, 6: Fengjing, 7: Sujiang, 8: Wuzhishan, 9: Tibetan, 10: Meishan, 11: Erhualian, 12: Wild boars; (**B**). RIPs Insertion loci diagram in *BMPR1B*; and (**C**). Sequence alignment for different species in *BMPR1B*.

**Figure 2 life-12-01650-f002:**
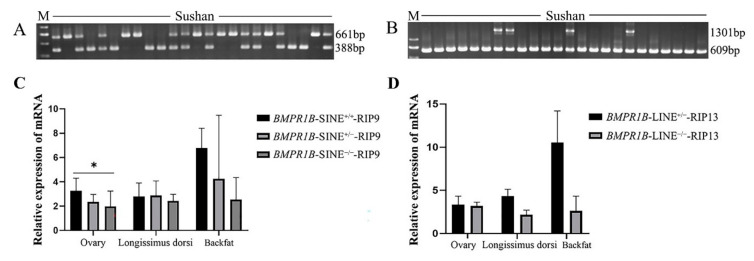
The expression pattern of pig *BMPR1B* in different genotype Sushan pigs of two RIPs. (**A**). Genotypes of *BMPR1B*-SINE-RIP9 in the Sushan pigs, M: DL2000 marker; (**B**). Genotypes of *BMPR1B*-LINE-RIP13 in the Sushan pigs, M: DL2000 marker; (**C**). Influence of SINE insertion on expression of *BMPR1B* gene; and (**D**). Effect of LINE insertion on expression of *BMPR1B* gene. The values are shown as mean ± SD, and * showed *p* < 0.05. “+/+” means Homozygous insert. “+/−” means heterozygous insert. “−/−” means no insert. The values are shown as mean ± SD, and * showed *p* < 0.05.

**Figure 3 life-12-01650-f003:**
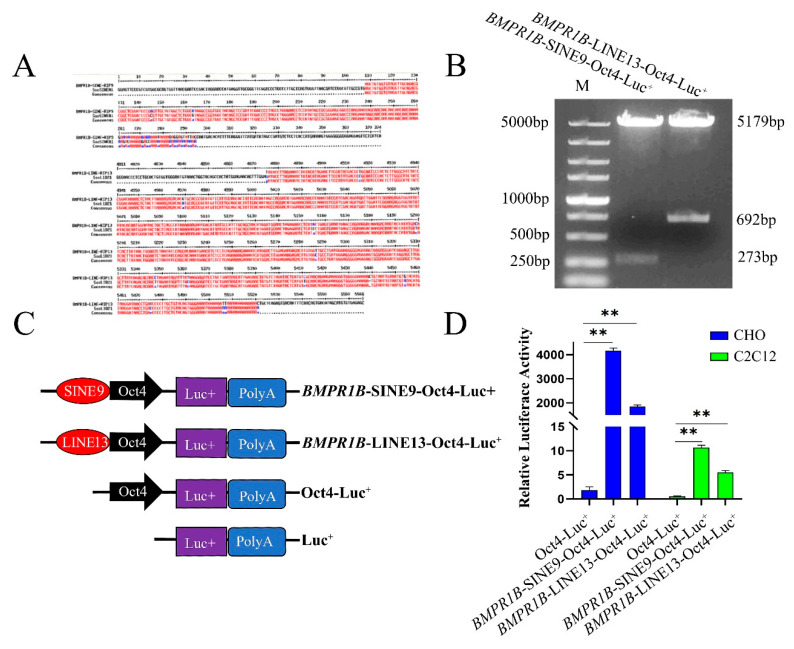
SINE and LINE insertion may act as a potential enhancer in CHO and C2C12 cells. (**A**). Result of target sequence and transposon reference sequence alignment; (**B**). The electrophoretogram of double-enzyme digestion for the recombinant vector, M: DL5000 marker; (**C**). A schematic diagram of the recombinant vector structure; and (**D**). A schematic diagram of the results of enhancer activity verification. The values shown are mean ± SD, and ** showed *p* < 0.01.

**Table 1 life-12-01650-t001:** The details of retrotransposon insertion in the *BMPR1B* gene.

Loci	Insertion in Exon/Intron	Length (bp)	Direction	Type	Chromosome (Sus scrofa11.1)
*BMPR1B*-SINE-RIP1	Intron1	286	−	SINEA1	125,030,204–125,030,205
*BMPR1B*-SINE-RIP2	Intron1	307	+	SINEA1	125,020,478–125,020,478
*BMPR1B*-SINE-RIP3	Intron1	259	−	SINEA3	124,975,543–124,975,544
*BMPR1B*-SINE-RIP4	Intron1	283	+	SINEA1	124,934,102–124,934,103
*BMPR1B*-SINE-RIP5	Intron2	304	+	SINEA1	124,833,678–124,833,679
*BMPR1B*-SINE-RIP6	Intron2	291	+	SINEA1	124,831,282–124,831,283
*BMPR1B*-SINE-RIP7	Intron2	285	−	SINEA1	124,829,563–124,829,564
*BMPR1B*-ERV-RIP8	Intron2	289	+	ERVIII	124,804,706–124,804,995
*BMPR1B*-SINE-RIP9	Intron2	273	−	SINEA1	124,800,210–124,800,483
*BMPR1B*-SINE-RIP10	Intron2	315	+	SINEA1	124,788,936–124,788,937
*BMPR1B*-LINE-RIP11	Intron5	134	−	L1D11	124,607,615–124,607,749
*BMPR1B*-ERV-RIP12	Intron5	565	−	ERVIII	124,606,427–124,606,428
*BMPR1B*-LINE-RIP13	Intron5	692	+	L1D21	124,595,110–124,595,802

Note: “+” indicates forward insertion, “−” indicates reverse insertion.

**Table 2 life-12-01650-t002:** Correlation between *BMPR1B*-SINE-RIP9, *BMPR1B*-LINE-RIP13 and growth traits of Large White pigs.

RIPs	Genotype	Number	Thickness of Back Fat/mm	Age at 30 kg Body Weight/d	Age at 100 kg Body Weight/d
*BMPR1B*-SINE-RIP9	SINE^+\+^	168	10.51 ± 0.20	74.99 ± 0.71	162.14 ± 0.55 ^a^
	SINE^+\^^−^	286	10.87 ± 0.17	74.91 ± 0.52	163.71 ± 0.52 ^A^
	SINE^−\^^−^	75	11.25 ± 0.34	74.89 ± 1.27	160.80 ± 0.90 ^b^
*BMPR1B*-LINE-RIP13	LINE^+/+^	259	10.65 ± 0.17	74.92 ± 0.52 ^a^	163.25 ± 0.56 ^a^
	LINE^+/−^	115	10.67 ± 0.28	74.36 ± 0.94 ^a^	161.97 ± 0.76 ^ab^
	LINE^−/−^	21	11.62 ± 0.62	70.07 ± 2.58 ^b^	159.26 ± 1.48 ^b^

Note: +/+: homozygote with retrotransposon insertion; +/−: heterozygote with retrotransposon insertion; −/−: homozygote without retrotransposon insertion. The same superscript letters in the same column (a and ab or b and ab) mean that the difference between groups was not significant. Different superscript lowercase letters (a and b) indicated a significant difference between groups (*p* < 0.05). Different superscript capital letters (A and b) indicated an extremely significant difference between groups (*p* < 0.01).

**Table 3 life-12-01650-t003:** Correlation between *BMPR1B*-SINE-RIP9 and reproductive performance of Large White pigs.

Genotype	Number	First Litter Size	Number Born Alive	Born Weight/kg
SINE^+/+^	99	10.47 ± 0.28 ^a^	9.86 ± 0.26	13.42 ± 0.36
SINE^+/−^	118	10.03 ± 0.23 ^ab^	9.49 ± 0.24	12.73 ± 0.33
SINE^−/−^	26	9.19 ± 0.57 ^b^	8.88 ± 0.63	11.85 ± 0.90

Note: +/+: homozygote with retrotransposon insertion; +/−: heterozygote with retrotransposon insertion; −/−: homozygote without retrotransposon insertion. The same superscript letters in the same column (a and ab or b and ab) mean that the difference between groups was not significant. Different superscript lowercase letters (a and b) indicated a significant difference between groups (*p* < 0.05).

**Table 4 life-12-01650-t004:** Distribution of RIPs in different pig breeds.

Polymorphic Site	Breeds	Number	Genotype Frequency			Allele Frequency		Hardy-Weinberg Equilibrium	Polymorphic Information Content
			+/+	+/−	−/−	+	−		
*BMPR1B*-SINE-RIP9	Large White	529	31.76	54.06	14.18	58.79	41.21	0.01	0.367
	Landrace	24	50.00	37.50	12.50	68.75	31.25	0.53	0.337
	Sushan	24	33.33	41.67	25.00	54.17	45.83	0.43	0.373
	Sujiang	21	80.95	19.05	0.00	90.48	9.52	0.63	0.157
	Wuzhishan	24	4.17	20.83	75.00	14.58	85.42	0.42	0.218
*BMPR1B*-LINE-RIP13	Large White	395	66.57	29.11	5.32	80.13	19.87	0.09	0.268
	Duroc	24	62.50	29.17	8.33	77.08	22.92	0.39	0.291
	Landrace	24	20.83	37.50	41.67	39.58	60.42	0.29	0.364
	Sushan	32	0	15.63	84.38	7.81	92.19	0.63	0.134
	Sujiang	21	4.76	19.05	76.19	14.29	85.71	0.31	0.215
	Jinhua	24	0	16.67	83.33	8.33	91.67	0.66	0.141
	Erhualian	24	0	4.17	95.83	2.08	97.92	0.92	0.040

Note: +/+: homozygote with retrotransposon insertion; +/−: heterozygote with retrotransposon insertion; −/−: homozygote without retrotransposon insertion; +: allele with retrotransposon insertion; and −: allele without retrotransposon insertion.

## Data Availability

The data are available from the corresponding author.

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
