# Peer review of "Two Retrotransposon Elements in Intron of Porcine BMPR1B Is Associated with Phenotypic Variation"

_life, 2022, doi:10.3390/life12101650_

Round 1

Reviewer 1 Report

Nil

Author Response

Thank you to our affirmation

Reviewer 2 Report

The study is interesting and significant. But it does not reached to publish, major revision should be given.

1.     Figure 2, in the pattern of agarose gel electrophoresis, detailed information should be given including tissues, fragment name, length, etc. Is the electrophoresis pattern of GAPDH given. It seems that panel B was the result of GAPDH electrophoresis, but it is not reasonable.

2.     Where is the tables?

3.     How relative luciferase activity were calculated, please given in the materials and methods.

Reviewer 3 Report

• lines 256-261: the description of the results in the text does not appear to agree with the results shown in Fig. 2. If the authors say the observed differences have a probability >0.05 of being due to chance they should say that, but Fig 2B clearly indicates that the LINE insertion clearly stimulates BMPR1B expression in the longissimus dorsi and backfat tissues.

• line 350: “obvious distribution pattern” should be spelled out to explain what is being distributed and how it is distributed.

• Table 4: why is the Hardy-Weinberg equilibrium number for Large White pigs so low when the allele frequencies are of comparable magnitudes. The authors need to explain how Polymorphic Information Content was calculated.

Round 2

Reviewer 2 Report

it is better to improve English language and style.